# Effects of Exercise in Improving Cardiometabolic Risk Factors in Overweight Children: A Systematic Review and Meta-Analysis

**DOI:** 10.3390/healthcare10010082

**Published:** 2022-01-01

**Authors:** Stefan Sebastian Busnatu, Liviu Ionut Serbanoiu, Andreea Elena Lacraru, Catalina Liliana Andrei, Cosmina Elena Jercalau, Marilena Stoian, Anca Stoian

**Affiliations:** 1Department of Cardiology, University of Medicine and Pharmacy “Carol Davila”, Emergency Hospital “Bagdasar-Arseni”, 050474 Bucharest, Romania; stefan.busnatu@umfcd.ro (S.S.B.); andreea.lacraru@umfcd.ro (A.E.L.); catalina.andrei@umfcd.ro (C.L.A.); cosmina-elena.jercalau@rez.umfcd.ro (C.E.J.); marinela.stoian@umfcd.ro (M.S.); anca.stoian@umfcd.ro (A.S.); 2Department of Diabetes, Nutrition and Metabolic Diseases, Carol Davila University of Medicine and Pharmacy, 020021 Bucharest, Romania

**Keywords:** physical stress, cardiometabolic, risk factors, long term intervention

## Abstract

This meta-analysis aims to evaluate the effects of exercise in improving cardiometabolic risk factors in overweight children and adolescents until the adolescent age, which is 18 years. A systemic search was conducted using the electronic databases PubMed/Medline, Cochrane Library, and Google Scholar, from inception to 29 June 2021. All statistical analyses were conducted in Review Manager 5.4.1. All studies meeting the inclusion criteria were selected. A random-effect model was used to pool the studies, and the results are reported in the odds ratio (OR) and corresponding 95% Confidence interval (CI). Twelve randomized control trials were selected for meta-analysis. Significant results were obtained for BMI in children after the interventions (0.38 95% CI 0.14, 0.62; *p* = 0.002; I^2^ = 65%). LDL level was also found significantly reduced (0.41 95% CI 0.01, 0.82; *p* = 0.05; I^2^ = 83%). Other factors such as HDL level, blood pressure, blood glucose level, body weight, and waist circumference were also analyzed. We found that exercise interventions significantly improved several cardiometabolic risk factors such as BMI, LDL level, BP, and blood glucose level. However, no significant effect on HDL concentration, waist circumference, and body weight were found. Long-term interventions are needed to attain improvement in all cardiometabolic risk factors.

## 1. Introduction

In the past 50 years, pediatric obesity (PO) has reached prevalence and severity levels worldwide, resulting in adverse effects on physical and mental health [1,2,3]. Furthermore, there is a significantly increased risk of developing cardiometabolic diseases, such as diabetes mellitus type 2, hypertension, coronary artery disease, and stroke as a consequence of pediatric obesity [4,5]. A high percentage of visceral abdominal fat is demonstrated to exacerbate hyperlipidemia and hypertension and contribute significantly to insulin resistance [6,7,8]. An extensive capillary network surrounds the adipose tissue, resulting in increased blood volume and cardiac output, affecting the cardiac workload [9,10]. The adipose tissue is also a significant source of interleukin-6 (IL-6), lipoprotein lipase, estrogen, angiotensinogen, adiponectin, leptin, insulin binding protein-3, and tumor necrosis factor- α [11,12]. IL-6 leads to a chronic inflammatory state that can trigger acute coronary syndrome. These properties of the adipose tissue increase the risk for cardiometabolic syndromes [13,14,15].

Fang et al., in their two-sample Mendelian randomization study, found a significant relationship between raised BMI and diabetes type 2 by analyzing the characteristics of nine single-nucleotide polymorphisms associated with increased risk of childhood obesity and increased risk of type 2 diabetes, glucose intolerance, insulin resistance, high blood pressure, and atherosclerosis [16]. In addition, obese children are often socially stigmatized and have psychological problems such as anxiety, depression, stress, low self-esteem, distorted body image, difficulties linked to being bullied, social withdrawal, and low quality of life, which results in poor school performance and lack of concentration [17,18,19,20].

Both genetic and environmental factors are found to influence PO. Pathological causes include hypothyroidism, Cushing syndrome, growth hormone deficiency, and central nervous system tumors such as craniopharyngioma [21,22,23,24,25]. Environmental factors such as consumption of fat-rich foods, lack of physical activity, and unhealthy lifestyle, biobehavioral and sociodemographic factors and mental health issues additionally contribute to cardiometabolic risk factors (CMRFs) [26]. Several gestational factors such as small or large gestational age, maternal smoking, maternal obesity, and Cesarean section are also found to increase the risk of childhood obesity [27,28]. Rare genetic defects in the leptin–melanocortin regulatory pathway result in obesity [16]. Body mass index (BMI) and the levels of high-density lipoprotein (HDL) and low-density lipoprotein (LDL) are widely used as strong and independent predictors of cardiovascular events [29,30,31].

Multidisciplinary interventions are considered effective in reducing CMRFs in overweight children and adolescents (overweight being defined as BMI between 25 and 30). These approaches primarily focus on weight reduction and lifestyle modification [32,33,34,35]. In a systematic review published by Rajjo et al., the cardiometabolic outcomes were significantly reduced with weight and BMI reduction. It was found that a reduction in BMI by 1.6 kg/m^2^ was associated with a reduction by 10 mmHg of systolic blood pressure and by 16 mg/dL of triglycerides and with an increase by 1.7 mg/dL of HDL [36]. The core elements of these programs include exercise and diet; other interventions such as pharmacological medications, psychotherapy, and surgery are less preferred [33,34]. A meta-analysis in 2019 concluded that exercise interventions in overweight children improve body composition by lowering body fat, blood glucose, BMI, and waist circumference [37]. High-intensity interval training (HIIT) is a popular multidisciplinary approach. HIIT was found to significantly improve aerobic capacity levels in overweight youth [38,39,40].

However, despite several previous studies, the results were not consistent or were inconclusive regarding the role of multidisciplinary interventions on HDL, LDL, and blood glucose levels in children. Our aim and objective for conducting this meta-analysis were to evaluate the effectiveness of exercise in reducing body weight, BMI, diastolic blood pressure (DBP), systolic blood pressure (SBP), HDL, LDL, and blood glucose, and waist circumference in overweight children and adolescents.

## 2. Materials and Methods

### 2.1. Data Sources and Search Strategy

This systematic literature search was conducted according to the Preferred Reporting items for Systematic Review and Meta-analyses (PRISMA) guidelines [41]. An electronic investigation from PubMed/Medline, Cochrane Library, and Google Scholar was conducted from their inception to 29 June 2021 (detailed strategy provided in Table 1), using the search string: ((physical exercise OR workout OR physical exertion) AND (cardiometabolic)) AND (risk factors OR predisposing Factor OR predictive factors) AND (children OR young OR adolescent OR youth OR teenagers). In addition, we manually screened the articles cited in previous meta-analyses, randomized controlled trials, cohort trials, and review articles to select all relevant studies.

### 2.2. Study Selection

All studies were included if they met the following eligibility criteria described as PICOS: (a) P (Participants) children and adolescents aged less than 18 years who were overweight; (b) I (Intervention) exercise or any other intervention leading to physical stress; C (Control) baseline values of the participants were considered as controls; (c) O (Outcome) effect of physical factors on cardiometabolic factors (body weight, BMI, diastolic blood pressure, systolic blood pressure, high-density lipoprotein, low-density lipoprotein, and blood glucose levels, and waist circumference); (d) S (Studies) articles with more than 5 participants who were overweight, with full follow-ups, excluding editorials, animal studies, observational studies, review articles, case reports, and case series.

Prisma checklist is added at the end of the study.

### 2.3. Statistical Analysis

Review Manager (RevMan) [Computer program]. Version 5.4. The Cochrane Collaboration, 2020, was used for all statistical analyses. The data from studies were pooled using a random-effects model to calculate the odds ratio (OR) with respective 95% confidence intervals (CI). Odds Ratio was used to describe the statistical probability of how physical stress affect cardiometabolic factors. The chi square test was performed to assess any differences between the subgroups. Sensitivity analysis was done to see if any individual study drove the results and explore reasons for high heterogeneity. As per the Cochrane handbook, the scale for heterogeneity was as follows: I^2^ = 25–60%—moderate; 50–90%—substantial; 75–100%—considerable heterogeneity; *p* < 0.1 indicated significant heterogeneity [42]. A *p* < 0.05 was considered significant for all analyses.

### 2.4. Data Extraction and Quality Assessment of the Studies

An independent search of the electronic databases was done. Studies searched were exported to the EndNoteTM 20.0.1 (Clarivate Analytics, Philadelphia, PA, USA), and duplicates were screened and removed. 

Data extraction and quality assessment of the included studies were done simultaneously. The Physiotherapy Evidence Database (PEDro) scale was used, which is known as a valid and reliable instrument to assess allocation to groups, blinding of allocation, and comparison between groups at baseline and its outcomes. This scale includes 11 questions with yes or no answers (yes = 1; no = 0), providing a total score which ranges between 0 (poor methodological quality) and 10 (excellent methodological quality; the first item is not included in the rating)). It suggests that the “level of evidence” be inserted. Based on the physiotherapy evidence database scale and in order to assess the evidence of the interventions, the Van Tulder criteria should be applied, according to which the selected studies were grouped by levels of evidence, on the basis of their methodological quality. A study with a physiotherapy evidence database score of 6 or more is considered level 1 (high methodological quality) (6–8: good, 9–10: excellent), while a study with a score of 5 or less is considered level 2 (low methodological quality)) (4–5: moderate; <4: poor). Details of quality assessment are provided in Table 2.

## 3. Results

### 3.1. Literature Search Results

The initial search of the three electronic databases yielded 884 potential studies. After exclusions based on titles and abstracts, the full text of 161 studies was read for possible inclusion. A total of 12 studies were selected for quantitative analysis [43,44,45,46,47,48,49,50,51,52,53,54]. Figure 1 summarizes the results of our literature search.

### 3.2. Study Characteristics

Table 3 and Table 4 provides the basic characteristics of the included studies and a description of the interventions used. Twelve studies had a total of 479 participants. The baseline values of the participants were considered as control values, and the values of factors after intervention were considered as experimental values. We analyzed eight factors: Body weight, BMI, Diastolic blood pressure, Systolic blood pressure, High-density lipoprotein, Low-density lipoprotein, Blood glucose, and Waist circumference (Table 5). In addition, a comparative analysis between the factors’ baseline and post-intervention values was carried out.

### 3.3. Publication Bias Assessment

As no factor was analyzed in more than 10 studies, no funnel plot is presented to show publication bias. 

### 3.4. Results of the Meta-Analysis

Detailed forest plots outlining the effect size of Body weight (Figure 2), Body mass index (Figure 3), Diastolic blood pressure (Figure 4), Systolic blood pressure (Figure 5), levels of High-density lipoprotein (Figure 6), Low-density lipoprotein (Figure 7), and Serum glucose (Figure 8), and Waist circumference (Figure 9) are provided in the manuscript.

#### 3.4.1. Body Weight

Seven out of the 12 studies reported data for Body weight. Pooled results (Figure 2) showed that Body weight was not statistically significant in the comparison with the control group (0.11 95% CI −0.05, 0.28; *p* = 0.18; I^2^ = 27%).

#### 3.4.2. Body Mass Index (BMI)

Out of 12 studies, 8 reported data for BMI. Pooled results (Figure 3) showed that BMI was statistically significant in the comparison with the control group (0.38 95% CI 0.14, 0.62; *p* = 0.002; I^2^ = 65%).

#### 3.4.3. Diastolic Blood Pressure

Out of 12 studies, 8 reported data for Diastolic blood pressure. Pooled results (Figure 4) showed that Diastolic blood pressure was statistically significant in the comparison with the control group (0.71 95% CI 0.11, 1.31; *p* = 0.02; I^2^ = 93%).

#### 3.4.4. Systolic Blood Pressure

Out of 12 studies, 8 reported data for Systolic blood pressure. Pooled results (Figure 5) showed that Systolic blood pressure was not statistically significant in the comparison with the control group (0.89 95% CI −0.02, 1.80; *p* = 0.05; I^2^ = 97%).

#### 3.4.5. High-Density Lipoprotein (HDL)

Out of 12 studies, 10 reported data for HDL. Pooled results (Figure 6) showed that HDL level was not statistically significant in the comparison with the control group (−0.48 95% CI −1.06, 0.11; *p* = 0.11; I^2^ = 92%).

#### 3.4.6. Low-Density Lipoprotein (LDL)

Out of 12 studies, 9 reported data for LDL. Pooled results (Figure 7) showed that LDL level was statistically significant in the comparison with the control group (0.41 95% CI 0.01, 0.82; *p* = 0.05; I^2^ = 83%).

#### 3.4.7. Blood Glucose

Out of 12 studies, 7 reported data for Blood glucose. Pooled results that (Figure 8) showed Blood glucose was statistically significant in the comparison with the control group (0.58 95% CI 0.13, 1.03; *p* = 0.01; I^2^ = 85%).

#### 3.4.8. Waist Circumference

Out of 12 studies, 8 reported data for Waist circumference. Pooled results (Figure 9) showed that Waist circumference was not statistically significant in the comparison with the control group (0.60 95% CI 0.02, 1.18; *p* = 0.04; I^2^ = 93%).

### 3.5. Sensitivity Analysis

A sensitivity analysis was conducted to assess the influence of each study on the overall effect by excluding one study at a time, followed by the generation of pooled Odds Ratio (OR) for the rest of the studies. No significant change was observed after the exclusion of any individual study, suggesting the results were robust.

## 4. Discussion 

In this study, we present the assessment of the evidence from 12 randomized control trials (*n* = 479), evaluating the effectiveness of exercise in reducing cardiometabolic risk factors in overweight children and adolescents from 6 to 18 years of age. During the intervention period from 5 weeks to 6 months, a reduction in BMI, blood glucose level, DBP, and LDL level was found. In contrast, there was no significant effect of exercise on body weight, SBP, HDL level, and waist circumference. HIIT intervention was used in four studies [44,46,49,53], moderate-intensity continuous training (MICT) was used as an intervention in four studies [44,48,52,53], and two studies used aerobic exercise [46,52]; the data regarding the type of exercise was not available in five studies [43,45,47,51,54]. Seabra et al. used school-based soccer practice as an intervention [50]. Children with metabolic syndrome were included by Kamal et al. [47].

In our literature review, we found that the first meta-analysis evaluating the effects of exercise and behavioral therapy on overweight children was published in 2012 and reported significant improvement after exercise in cardiometabolic outcomes compared with overweight children without treatment [55]. Several systematic reviews and meta-analyses have been published that evaluate the positive effects of HIIT on CMRFs and cardiorespiratory fitness [56,57]. School-based exercise programs had also a significant effect in decreasing CMRFs in a meta-analysis [58]. Several RCTs conducted on children with metabolic syndrome to evaluate the effectiveness of metformin showed significant results [59,60]. Various studies assessed the association of dietary patterns with obesity [61,62,63,64,65]. The association of sleeping patterns with obesity appeared evident [66,67,68]. A meta-analysis was published in 2016, which showed a significant decrease in BMI and improvement in body composition in overweight children after exercise [69]. However, the data regarding the effectiveness of exercise on HDL, LDL, and blood glucose levels were inconclusive in this study.

BMI is the most reliable and accessible screening tool in clinical settings to assess obesity and overweight in children and adolescents [70,71,72,73]. We found a significant effect of exercise in reducing BMI in children and adolescents [73]. BMI was reported in 8 studies, and the studies were further divided based on their intervention; therefore, 13 interventions were analyzed. Both aerobic and MICT interventions, examined in Sigal et al. [52], in two patients’ populations, as reported by Kamal et al., had a significant effect in reducing BMI [47]. On the other hand, nine interventions did not significantly affect BMI [43,44,45,46,48,53], as shown in Figure 3. The most prolonged intervention period was used by Sigal et al. and corresponded to 6 months. It resulted in a significant decrease in BMI, i.e., −0.6 (95% CI, −1.1 to 0) after aerobic training and −0.5 (95% CI, −1.1 to 0) after MICT [52]. Seabra et al. also used an intervention period of 6 months in their school-based soccer program and also found a significant decrease in BMI z-score in overweight children and adolescents (BMI z-score = 2.2 ± 0.5, %BFM = 32.3 ± 6.1) [50]. The intervention period of Kamal et al. was 12 weeks, and significant results were attained in patients with and without metabolic syndrome. However, better effects were obtained for children with metabolic syndrome. Metabolic syndrome decreased from 12.9% to 7.5%, and BMI was reduced from 47.3% to 32.6%, suggesting an improvement in metabolic syndrome with exercise [47]. An intervention period from 4 weeks to 12 weeks was used in the rest of the studies and showed non-significant results [43,44,45,46,48,53].

No significant change was found regarding body weight and waist circumference. Seven studies were analyzed reporting 12 interventions for body weight reduction; only one intervention had a significant effect [47]. Eight studies reported 11 waist circumference interventions; 4 had significant effects [45,47,52]. Figure 9 shows the data regarding body weight and waist circumference. For body weight, significant changes were only seen in children with metabolic syndrome after 12 weeks of intervention, as reported by Kamal et al. (standard mean difference (SMD) 1.23; 95% CI 0.34 to 2.11) [47]. Waist circumference was improved in the metabolic syndrome patients examined by Kamal et al. (SMD 1.31; 95% CI 0.41 to 2.20), Hobkrik et al. (SMD 0.48; 95% CI 0.15 to 0.8) and Sigal et al. with two interventions had; (SMD 1.82; 95% CI 1.44 to 2.19) and SMD 2.39; 95% CI 1.96 to 2.81) [45,47,52]. The results suggest that waist circumference, body weight, and BMI were improved in children and adolescents with metabolic syndrome. Long-term interventions also significantly affected these parameters.

The interactions between different genetic and environmental factors play a significant role in the pathogeny of cardiometabolic syndrome [74]. Increased LDL and blood glucose levels play an essential role in the manifestation of atherosclerosis, leading to coronary artery diseases (CAD) [75,76,77,78]. Overweight children and adolescents are more prone to develop adulthood CAD [79,80]. We found the exercise has a vital role in decreasing LDL and blood glucose levels in children. Nine studies using 11 interventions were analyzed to examine the effects on LDL. Four interventions indicated a positive effect of physical exercise [45,47,49,54], one intervention showed non-significant results [51]. Seven studies reported the blood glucose levels after 10 interventions; 4 interventions had significant effects [45,47,51,53] (Figure 8). Shalitin et al. showed a significant increase in LDL level after an intervention of 12 weeks (SMD −0.58; 95% CI −0.97 to −0.19); however, in a follow-up after 52 weeks, it was found that LDL level was significantly reduced (*p* = 0.004) [51]. The 12-week intervention of Shalitin et al. was significantly effective in reducing blood glucose level (SMD 2.09; 95% CI 1.61 to 2.57). Significant results were also obtained for children and adolescents affected by metabolic syndrome [47]. No significant effect on HDL was found after the analysis (Figure 6). For HDL, 13 interventions reported in 10 studies were analyzed; 5 interventions had no effect [46,47,54], and 2 interventions improved physical stress [45,51]. Children and adolescents with metabolic syndrome showed a significant increase in HDL concentration (SMD −2.30; 95% CI −3.37 to −1.23). Shalitin et al. showed a significant reduction in HDL levels after a 12-week intervention (SMD 1.75; 95% CI 1.30 to 2.21).

According to the World Health Organization (WHO), hypertension is the leading cause of death worldwide [81]. Weight loss shows a significant effect in reducing blood pressure [82,83]. Therefore, we emphasize the significance of exercise for blood pressure reduction. We found a significant decrease in diastolic blood pressure but not in systolic blood pressure (Figure 4 and Figure 5). Eight studies reported 11 interventions for blood pressure. Two interventions significantly reduced SBP [52], while 3 interventions significantly reduced DBP [45,52]. Sigal et al.’s 6-month intervention showed significant results in decreasing both SBP and DBP [52]. Hobkrik et al. showed a significant change in DBP but not in SBP (SMD 0.40; 95% CI 0.07 to 0.73).

Therefore, we conclude that long-term interventions can obtain significant results, regardless of the type of intervention. Furthermore, early effects were seen only in children and adolescents affected by metabolic syndrome.

## 5. Limitations

Our study is limited by the following factors: (a) all studies were randomized controlled trials in nature, (b) fewer studies were available with significant publication bias (the reason is described in the above section), (c) the number of study subjects was low, (d) studies reporting on different types of physical activity were pooled together, which might be the reason for the high heterogeneity observed. Nevertheless, these studies were pivotal for our analysis. More studies providing greater numbers of subjects and random controls should be conducted.

## 6. Conclusions

The current evidence suggests that exercise interventions in overweight children and adolescents improve BMI, DBP, and LDL and blood glucose levels; however, no significant effects were obtained for HDL level, body weight, SBP, and waist circumference, suggesting that exercise interventions are substantial in improving some cardiometabolic risk factors. However, long-term interventions might significantly improve other cardiometabolic risk factors. Therefore, the risks of cardiometabolic diseases in overweight adolescents can be reduced with a long-term exercise program and lifestyle modification.

## Figures and Tables

**Figure 1 healthcare-10-00082-f001:**
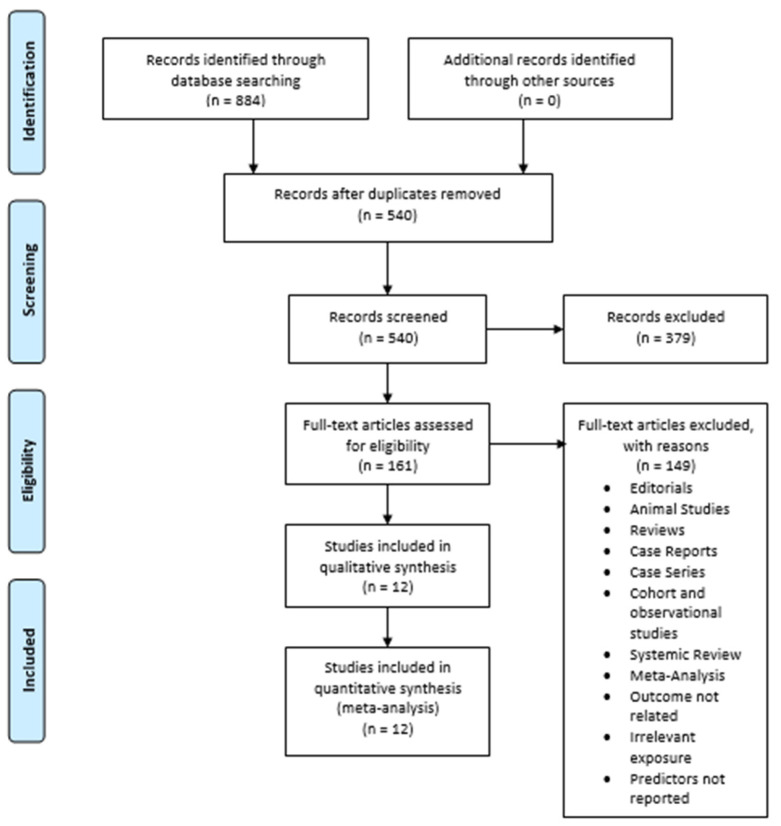
Prisma flow chart.

**Figure 2 healthcare-10-00082-f002:**
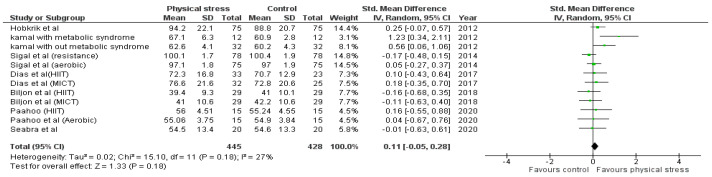
Forest plot showing the results for body weight.

**Figure 3 healthcare-10-00082-f003:**
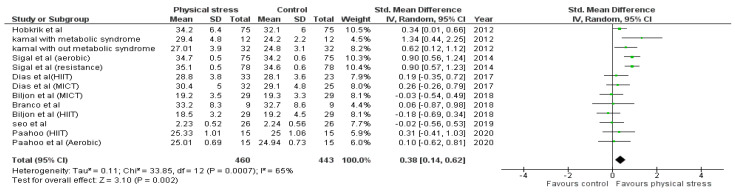
Forest plot showing the results for BMI.

**Figure 4 healthcare-10-00082-f004:**
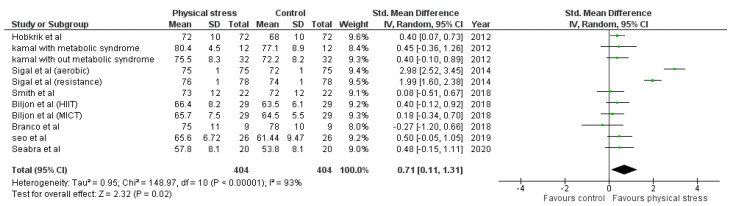
Forest plot showing the results for DBP.

**Figure 5 healthcare-10-00082-f005:**
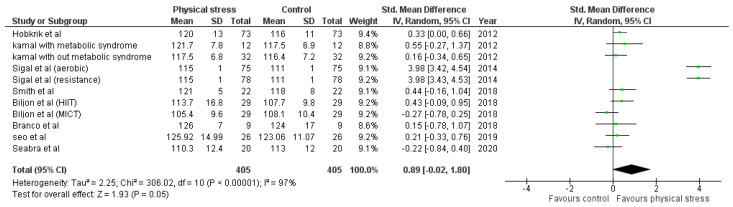
Forest plot showing the results for SBP.

**Figure 6 healthcare-10-00082-f006:**
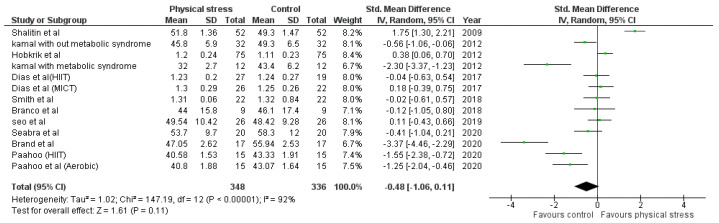
Forest plot showing the results for HDL.

**Figure 7 healthcare-10-00082-f007:**
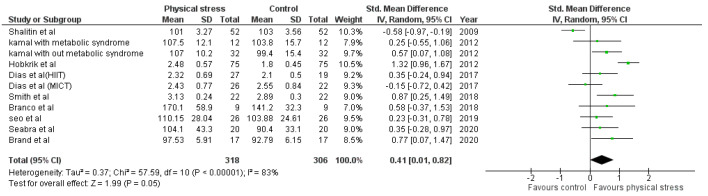
Forest plot showing result of LDL.

**Figure 8 healthcare-10-00082-f008:**
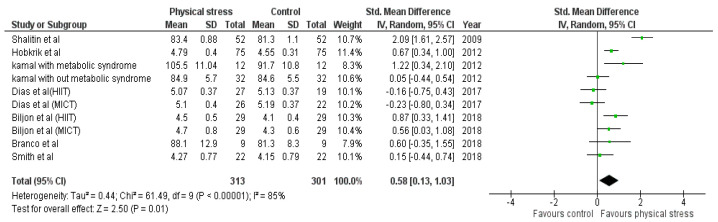
Forest plot showing the results for blood glucose.

**Figure 9 healthcare-10-00082-f009:**
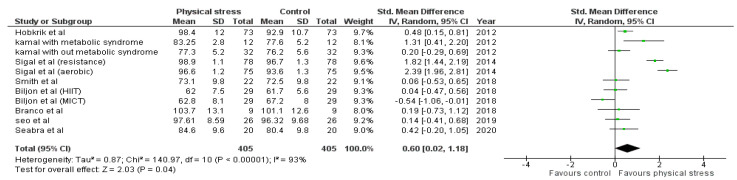
Forest plot showing the results for waist circumference.

**Table 1 healthcare-10-00082-t001:** Detailed search strategy.

Search Engine	Search Strategy
Pubmed/Medline	(“exercise”[MeSH Terms] OR “exercise”[All Fields] OR (“physical”[All Fields] AND “exercise”[All Fields]) OR “physical exercise”[All Fields] OR (“workout”[All Fields] OR “workouts”[All Fields]) OR (“physical exertion”[MeSH Terms] OR (“physical”[All Fields] AND “exertion”[All Fields]) OR “physical exertion”[All Fields])) AND (“cardiometabolic”[All Fields] OR “cardiometabolically”[All Fields]) AND (“risk factors”[MeSH Terms] OR (“risk”[All Fields] AND “factors”[All Fields]) OR “risk factors”[All Fields] OR (“causality”[MeSH Terms] OR “causality”[All Fields] OR (“predisposing”[All Fields] AND “factor”[All Fields]) OR “predisposing factor”[All Fields]) OR ((“predict”[All Fields] OR “predictabilities”[All Fields] OR “predictability”[All Fields] OR “predictable”[All Fields] OR “predictably”[All Fields] OR “predicted”[All Fields] OR “predicting”[All Fields] OR “prediction”[All Fields] OR “predictions”[All Fields] OR “predictive”[All Fields] OR “predictively”[All Fields] OR “predictiveness”[All Fields] OR “predictives”[All Fields] OR “predictivities”[All Fields] OR “predictivity”[All Fields] OR “predicts”[All Fields]) AND (“factor”[All Fields] OR “factor s”[All Fields] OR “factors”[All Fields]))) AND (“child”[MeSH Terms] OR “child”[All Fields] OR “children”[All Fields] OR “child s”[All Fields] OR “children s”[All Fields] OR “childrens”[All Fields] OR “childs”[All Fields] OR (“young”[All Fields] OR “youngs”[All Fields]) OR (“adolescences”[All Fields] OR “adolescency”[All Fields] OR “adolescent”[MeSH Terms] OR “adolescent”[All Fields] OR “adolescence”[All Fields] OR “adolescents”[All Fields] OR “adolescent s”[All Fields]) OR (“adolescent”[MeSH Terms] OR “adolescent”[All Fields] OR “youth”[All Fields] OR “youths”[All Fields] OR “youth s”[All Fields]) OR (“adolescent”[MeSH Terms] OR “adolescent”[All Fields] OR “teenage”[All Fields] OR “teenager”[All Fields] OR “teenagers”[All Fields] OR “teenaged”[All Fields] OR “teenager s”[All Fields] OR “teenages”[All Fields]))
Cochrane	((physical exercise OR workout OR physical exertion) AND (cardiometabolic)) AND (risk factors OR predisposing Factor OR predictive factors) AND (children OR young OR adolescent OR youth OR teenagers)
Google Scholar	((physical exercise OR workout OR physical exertion) AND (cardiometabolic)) AND (risk factors OR predisposing Factor OR predictive factors) AND (children OR young OR adolescent OR youth OR teenagers)

**Table 2 healthcare-10-00082-t002:** Quality assessment of the selected articles (PEDro Scale).

Authors	1	2	3	4	5	6	7	8	9	10	11	Total Score	Methodological Quality
Seo et al. [43]	Y	1	0	1	1	1	0	1	1	1	1	8	High
Dias et al. [44]	Y	1	1	1	1	1	1	1	1	1	1	10	High
Hobkrik et al. [45]	Y	1	1	1	1	1	1	1	1	1	1	10	High
Paahoo et al. [46]	Y	1	1	1	1	1	1	1	1	1	1	10	High
Kamal et al. [47]	Y	0	0	0	0	0	0	1	1	1	1	4	Low
Branco et al. [48]	Y	1	1	1	1	1	1	1	1	1	1	10	High
Smith et al. [49]	Y	1	1	1	1	1	1	1	1	1	1	10	High
Seabra et al. [50]	Y	0	0	0	0	0	0	1	1	1	1	4	Low
Shalitin et al. [51]	Y	1	1	1	1	1	1	1	1	1	1	10	High

Abbreviations: 1, Eligibility; 2, Random allocation; 3, Concealed allocation; 4, Baseline comparability; 5, Blind subjects; 6, Blind therapists; 7, Blind assessors; 8, Adequate follow-up; 9, Intention-to-treat analysis; 10, Between-group comparisons; 11, Point estimates and variability; Y, yes; N, No. Note: Eligibility criteria item does not contribute to the total score.

**Table 3 healthcare-10-00082-t003:** Characteristics of the included studies.

Author	Country	Total Sample by Gender (*n*)	Age (Years) (M ± SD)	Groups (*n*)	Body Fat (%)	Body Mass (kg)	BMI (kg/m^2^)	Registered Protocol
Seo et al., 2019 [43]	Korea	25F	CG = 12.09 ± 2.20	CG = 44	CG = 41.26 ± 4.25	CG = 72.1 ± 19.88	CG = 29.43 ± 4.81	Yes
			EG = 12.8 ± 1.72	EG = 26	EG = 41.77 ± 4.23	EG = 77.4 ± 11.34	EG = 30.06 ± 3.40	
Dias et al. 2017 [44]	Australia	103F	CG = 11.5 ± 2.4	CG = 100	CG = 44.1 ± 6.2	CG = 31.3 ± 10.6	CG = 29.5 ± 4.4	Yes
			EG = 12.0 ± 2.3	EG = 99	EG = 19.5 ± 7.5	EG = 7.7 ± 3.6	EG = 17.6 ± 2.1	
Hobkrik et al., 2016 [45]	United Kingdom	N/R	N/R	75	N/R	94.2 ± 22.1	34. 2 ± 6.4	N/R
Pahoo et al., 2020 [46]	Iran	N/R	CG = 11.20 ± 0.94	CG = 15	CG = 27.88 ± 1.06	CG = 54.20 ± 4.45	CG = 25.02 ± 1.89	N/R
			HIITG = 11.13 ± 0.99	HIITG = 15	HIITG = 28.04 ± 1.46	HIITG = 56.00 ± 4.51	HIITG = 25.33 ± 1.01	
			AG = 10.86 ± 1.06	AG = 15	AG = 27.87 ± 1.06	AG = 55.06 ± 3.75	AG = 25.01 ± 0.69	
Kamal et al., 2012 [47]	Egypt	40F	CG = 10.1 ± 1.21	CG = 49	N/R	CG = 62.6 ± 4.1	CG = 67.1 ± 6.3	N/R
			EG without MS = 10.2 ± 1.2	EG without MS = 32		EG without MS = 50.2 ± 3.7	EG without MS = 17.2 ± 2.5	
			EG with MS = 11.04 ± 1.15	EG with MS = 12		EG with MS = 67.1 ± 6.3	EG with MS = 29.4 ± 4.8	
Branco et al., 2016 [48]	Brazil	N/R	FG = 16 ± 1	FG = 9	FG = 43.9 ± 4	FG = 99 ± 20.5	FG = 34.7 ± 3.8	Yes
			WG = 16 ± 1	WG = 9	WG = 38.7 ± 9.2	WG = 97.8 ± 24.2	WG = 33.2 ± 8.3	
Smith et al., 2017 [49]	United Kingdom	20F	CG = 16.8 ± 0.5	CG = 30	N/R	CG = 66.2 ± 13.8	CG = 21.8 ± 2.1	N/R
			EG = 17 ± 0.3	EG = 22		EG = 67.1 ± 14.4	EG = 22.5 ± 2.5	
Seabra et al., 2020 [50]	Germany	40M	CG = 10.1 ± 1.5	CG = 20	CG = 35.1 ± 8.3	CG = 57.6 ± 15.7	N/R	N/R
			EG = 10.5 ± 1.5	EG = 20	EG = 34.4 ± 6	EG = 54.5 ± 13.4		
Shalitin et al., 2009 [51]	Israel	81F	EG = 8.21 ± 1.78	EG = 52	EG = 41.2 ± 1.36	EG = 46 ± 1.6	EG = 25.5 ± 0.52	N/R
			D + EG = 8.2 ± 1.56	D + EG = 55	D + EG = 41.3 ± 1.3	D + EG = 46.4 ± 1.51	D + EG = 25.9 ± 0.51	
			DG = 8.51 ± 1.52	DG = 55	DG = 43.1 ± 1.24	DG = 47.5 ± 1.52	DG = 25.5 ± 0.52	
Sigal et al., 2014 [52]	Canada	213F	AG = 15.5 ± 1.4	AG = 75	AG = 47.1 ± 1.3	AG = 97.1 ± 1.8	AG = 34.7 ± 0.5	N/R
			RG = 15.9 ± 1.5	RG = 78	RG = 48 ± 1.3	RG = 100.1 ± 1.7	RG = 35.1 ± 0.5	
			CTG = 15.5 ± 1.3	CTG = 75	CTG = 48.4 ± 1.3	CTG = 97.8 ± 1.8	CTG = 34.7 ± 0.5	
			CG = 15.6 ± 1.3	CG = 76	CG = 46.6 ± 1.3	CG = 97.9 ± 1.8	CG = 34.2 ± 0.5	
Biljon et al., 2018 [53]	South Africa	67F	11.1 ± 0.8	MICTG = 29	N/R	MICTG = 41.0 ± 10.6	MICTG = 19.2 ± 3.5	N/R
				HIITG = 29		HIITG = 41.0 ± 10.6	HIITG = 18.5 ± 3.2	
				MICTG + HIITG = 27		MICTG + HIITG = 38.7 ± 9.3	MICTG + HIITG = 18.0 ± 3.8	
				CG = 24		CG = 42.6 ± 9.6	CG = 20.3 ± 3.7	
Brand et al., 2020 [54]	Brazil	22F	CG = 8.27	CG = 18	CG = 14.13	CG = 39.29	CG = 21.88	Yes
			EG = 8.17	EG = 17	EG = 13.6	EG = 40.9	EG = 21.97	

Abbreviations: N, number; M, mean; SD, standard deviation; CG, control group; EG, experimental group; F, female; M, male; USA, united states; BMI, body mass index; kg, kilograms; NR, not reported; HIITG, high-intensity interval training group; AG, aerobic group; MS, metabolic syndrome; FG, functional group; WG, weight training group; D, diet; DG, diet group; CTG, combined training group MIITG, moderate intensity interval training group.

**Table 4 healthcare-10-00082-t004:** Description of the interventions performed in the included studies.

Author	Exercise Modality	Exercise (Names)	Frequency (Days/Week)	Intensity	Sets/Exercise (*n*)	Reps per Set (*n*)	Rest	Intervention Duration (Weeks)	Session Duration (min)	Eccentric Velocity (s)	Supervised?
Seo et al., 2019 [43]	Cardio training	Aerobic exercise and ICAAN exercise	3	moderate-intensity intervention	N/R	N/R	30–40 s	16	60	N/R	Yes
Dias et al. [44]	Cardio training	treadmill	3	High- and Moderate-Intensity Interval Training	N/R	N/R	3 min	12 weeks	36 (High intensity) 44 (Moderate intensity)	N/R	Yes
Hobkrik et al. [45]	Cardio training	N/R	N/R	N/R	N/R	N/R	N/A	4	N/R	N/R	Yes
Pahoo et al. [46]	Cardio training and Flexibility	warm-up and other exercise	3	High-Intensity Interval Training and Aerobic Exercise	3	N/R	5 min	12	45	N/R	Yes
Kamal et al. [47]	Cardio training	warm-up, walking–jogging, and relaxation exercises	3	N/R	N/R	N/R	N/R	12 weeks	30–65	N/R	Yes
Branco et al. [48]	Cardio training and Strength training	warm-up, CP, HS, SP, Leg extension, Triceps pulley, ACM, OAM, Scott curl machine, Aerobics, CRM, Incline row, Leg curl, LCGP, Push-ups, SWS, MBV, GUD, Triceps bench dips, SSU, HLKF, RTRX, TPNR, BTRX, OSSB	3	High- and Moderate- Intensity Interval Training	3	N/R	3 min	12 weeks	46	N/R	
Smith et al. [49]	Cardio training	“all out” running sprints	3	Sprint Interval Training	5–6	N/R	30 s	4 weeks	4.5–5.5	N/R	Yes
Seabra et al. [50]	Cardio training	soccer	2	N/R	N/R	N/R	10 min	6 months	60–90	N/R	Yes
Shalitin et al. [51]	Cardio training and Strength training	Sports, running games, sit-ups, hand-lifting of small weights, and ball exercise	3	Aerobic exercise and resistance training exercises	N/R	N/R	N/R	12 weeks	45 (each group)	N/R	Yes
Sigal et al. [52]	Cardio training and Strength training	Gymnasiums, weight machines or free weights	2–4	aerobic and resistance training	2–3	8–15	N/R	6 months	20–45	N/R	Yes
Biljon et al. [53]	Cardio training	Warm-up and cool-down periods consisting of jogging at a low intensity, followed by static stretching	3	High- and Moderate- Intensity Interval Training	N/R	N/R	5 min	5 weeks	23 (High intensity) 33 (Moderate intensity)	N/R	Yes
Brand et al. [54]	Multicomponent	Exercise	2	N/R	N/R	N/R	N/R	12 weeks	60	N/R	Yes

Abbreviations: N, number; M, mean; SD, standard deviation; CG, control group; EG, experimental group; F, female; M, male; BMI, body mass index; kg, kilograms; NR, not reported. CP, Chest press; SP, Shoulder press; HS. Hack squat; ACM, Abdominal crunch machine; OAM, Oblique abdominal machine; CRM, Calf raise machine; LCGP, Lever close grip pulldown; SWS, Squat with Swiss-ball on the back; MBV, Medicine ball vertical throw; GUD, Go up and down a plinth; SSU, Straight sit-ups on ball; HLKF, Hip lift with knee flexion on Swiss-ball; RTRX, Row on TRX with supinated grip; BTRX, Biceps curl on TRX; OSSB, Oblique sit-ups on Swiss-ball; TPNR, Tire pulling with naval rope.

**Table 5 healthcare-10-00082-t005:** Details about the selected factors.

Factors	No. of Studies	Odds Ratio (OR 95% CI)	*p*-Value	Heterogeneity (*I^2^)*
	(*n*)			(%)
Body weight	7	0.11 95% CI −0.05, 0.28	0.18	27
BMI	8	0.38 95% CI 0.14, 0.62	0.002	65
Diastolic blood pressure	8	0.71 95% CI 0.11, 1.31	0.02	93
Systolic blood pressure	8	0.89 95% CI −0.02, 1.80	0.05	97
High-density lipoprotein	10	−0.48 95% CI −1.06, 0.11	0.11	92
Low-density lipoprotein	9	0.41 95% CI 0.01, 0.82	0.05	83
Blood glucose	7	0.58 95% CI 0.13, 1.03	0.01	85
Waist circumference	8	0.60 95% CI 0.02, 1.18	0.04	93

## Data Availability

Not applicable.

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
