# Peer review of "Effects of Exercise in Improving Cardiometabolic Risk Factors in Overweight Children: A Systematic Review and Meta-Analysis"

_healthcare, 2022, doi:10.3390/healthcare10010082_

Round 1

Reviewer 1 Report

Comments are included in the attached document.
Kind regards

Author Response

Firstly, I would like to show my gratitude for taking time from you schedule and reviewing our manuscript. We are happy to see that you found our manuscript interesting. We managed to change the structure of the references

Reviewer 2 Report

In “Effects of exercise in improving the cardiometabolic risk factors 3 in overweight children: a systematic review and meta-analysis”, the authors evaluated the effectiveness of exercise on obese adolescents in reducing cardiometabolic risk factors, which included body weight, BMI, diastolic blood pressure, and systolic blood pressure, HDL, LDL, blood glucose, and waist circumference. Although systematically exploring the effects of pediatric exercise interventions metabolic risk factors in “at-risk” population may be warranted, the authors did not make a compelling case for why this particularly systematic review was needed, and how it differed or added to our current knowledge. Additionally, although it was noted that the PRISMA guidelines were followed, clarity within many of the PRISMA checklist item is needed for transparent reporting of the approach.

BROAD COMMENTS:

Overall, the introduction did not motivate the need for a review. It is not clear what this is adding to our knowledge/the literature.

Additional, and pivotal information is needed in the search strategy to be able to understand (or even replicate) the approach/search. Justification for decisions would be helpful, but more importantly, it is possible that many articles were missed with the key terms used here.

The title suggests that “overweight” children are the focus of this review, but often throughout the manuscript, participants are referred to as obese. Overweight is not necessarily exchangeable with obese, and I don’t believe either were defined in the context of this review.  

SPECIFIC COMMENTS

Introduction

Line 31: The authors may want to clarify this sentence if they are referring to increased risk in obese children (vs in general).

Line 40: This paragraph could use a topic/transition sentence.

Line 48: Have only genetic and feeding behaviors been linked? What about movement/sleep behaviors/habits?

Line 48: It appears that "Pathological causes" should be the start of a new sentence?

Line 51: Please see note for Line 48 - suggest revising that intro sentence slightly as physical activity is mentioned here.

Line 58: Suggest deleting "The". Also, could the authors explain what is meant by "multidisciplinary" - multiple intervention components/settings/approaches?

Lines 59-64: This paragraph opens up by stating multidisciplinary programs were effective, but then exercise interventions are described as examples. This introduction could benefit from a transition a broad discussion of potential causal factors of obesity in children and effective interventions to why it is important to systematically explore to effects of exercise interventions in obese children. Currently, it feels some important information is missing to motivate the aims of this review. Also, what is known from other reviews and why is an additional one needed?

Line 65: How do the authors define "cardiometabolic risk factors"? Although BMI and body weight are risk factors of both obesity and obesity-related diseases, should they be considered "cardiometabolic factors"? Perhaps, in alignment with other public health reviews, "indictors of adiposity", particularly as they are not direct measurements of body composition.

Line 66: Did this study focus on children or adolescents? Terms seem to differ throughout manuscript. 

Methods

Line 72: It would be helpful for the authors to include a PRISMA checklist here as well. Although most components are touched on here in the methods, greater detail in some areas is important for transparency of the approach. Many programs in children may be phrased as a physical activity program/intervention and therefore would not appear in this search. This search as described could have excluded some key studies. (In fact 12 seems rather low, particularly as BMI and weight were included as outcomes.) I did a quick search in PubMed and just used "overweight children" and "exercise" and "blood pressure" and got 670 results and found some potential articles that were not included here. For example:

Staiano AE, Beyl RA, Guan W, Hendrick CA, Hsia DS, Newton RL Jr. Home-based exergaming among children with overweight and obesity: a randomized clinical trial. Pediatr Obes. 2018 Nov;13(11):724-733. doi: 10.1111/ijpo.12438. Epub 2018 Jul 20. PMID: 30027607; PMCID: PMC6203598.

van Leeuwen J, Andrinopoulou ER, Hamoen M, Paulis WD, van Teeffelen J, Kornelisse K, van der Wijst-Ligthart K, Koes BW, van Middelkoop M. The effect of a multidisciplinary intervention program for overweight and obese children on cardiorespiratory fitness and blood pressure. Fam Pract. 2019 Mar 20;36(2):147-153. doi: 10.1093/fampra/cmy061. PMID: 29939242.

Line 83: All ages or children only? This is very important to note. Also, what were considered cardiometabolic factors? Please be specific so this search strategy could be replicated.

Line 84: Could this be clarified? Articles had to report sample sizes or there was a minimum sample size required? Did all children in a program have to be overweight and if so, how was that defined? As it stands, there is not enough information, and some decisions may have impacted how "accurate" this search strategy was.

Line 87: Please be more specific in this section. What data/information was extracted? Table 2 doesn't appear to describe the quality assessment, rather it appears to report the results. What elements were assessed for quality and how were they rated? (e.g., what criteria was used to classify each category as "unclear risk", "low risk", "high risk")? How was overall quality determined for a study or overall for all studies (if at all)?

Line 89: Some additional explanation would be helpful here as well. What is the OR describing?

Results

Line 115-116: This is unclear.

Lines 118-119: What is meant by comparative study? This section could benefit from a little more clarity about what was assessed and how.

Table 3: A more detailed summary table should be included with specific information about the exercise program (dose, modality, length of program etc.), specific measures and timepoints.

Discussion

Line 196: This age range was not defined in your criteria in methods.

Line 207: The title says overweight children, not obese?

Line 219: I don't necessarily agree with this assessment. Although there could be some debate, it could be argued that BMI should be used only as a screening tool (particularly if taken at only one point in time), rather than a diagnostic tool in a medical/clinical setting.

Author Response

Firstly, I would like to show my gratitude for taking time from you schedule and reviewing our manuscript. We would like to appreciate you reviewing our manuscript and providing us useful insight to make our manuscript better.

For Introduction:

We have worked according to your line sequence and have changed and provide details regarding your queries. We have explained the word ‘multidisciplinary’ and what are the cardiometabolic factors we intend to work on. We have also specified the age and that we took overweight participants.

For Methods:

We have provided PRISMA checklist at the end of manuscript (after references) as directed by you. We assure you that we have not missed any articles as are inclusion/exclusion criteria with our search string allowed us to obtain all possible and relevant studies. Staiano et al must be excluded as it does not have follow-up mean ans standard deviation (it has baseline values and post intervention mean difference which we can’t use in our method of analysis). Leeuwena et al has used percentiles which is also not part of our method. We have updated our literature search criteria as per your instructions and we have stated the domains used by predefined criteria according to Cochrane Collaboration’s Tool. We have also described why we used Odds Ratio here.

For Results:

We have added details in the portion you highlighted for us and updated the table to provide the modalities of the given studies.

For Discussion:

We have added age criteria and stated that participants were overweight. We have also provided more discussion so that it becomes clearer about the importance of BMI in clinical practice.

For English:

We are happy to pay for the editing and modification of the English language in our manuscript

Reviewer 3 Report

The purpose of this systematic review and meta-analysis study was to evaluate the effects of exercise in improving the cardiometabolic risk factors in overweight children. The findings show that exercise intervention significantly improves several cardiometabolic risk factors such as BMI, LDL, BP, and blood glucose. However, no significant effect on HDL, waist circumference, and body weight were found.

Although I found the research interesting, I had the major concerns regarding several aspects.

  1. The significance of the manuscript is not very clear. The introduction focusses a lot on the harm of pediatric obesity (PO) and its causes (paragraph 1-3), but the effects of the exercise on PO (which is the focus of the current review) are not clearly stated.
  2. The findings of the present review (e.g., BMI, blood glucose, and waist circumference) are similar to the mentioned meta-analysis. Please explain more on the strengths of the present review compared to the previous meta-analyses.
  3. The heterogeneity level in the majority of outcomes is pretty high, which may undermine the validity of the present findings analysis if no further analysis was conducted to explore the reasons of such a high heterogeneity. Please provide details of sensitivity analyses or sub-group analysis preformed as mentioned in the “2.4. Statistical analysis”. Furthermore, meta-regression is also recommended to perform to enhance the importance of the manuscript.
  4. In Methods, “The systematic literature search was conducted according to the Preferred Reporting 71 items for Systematic Review and Meta-analyses (PRISMA) guidelines”, but why did not register with PROSPERO database? Or could you provide reference number if not stated?
  5. Of study selection (Line 82-86). The inclusion criteria should be more detailed. For example, the participants included should not only be “articles having patients undergoing physical exercise” but adolescents under 18 years old. Also, the types, intensity or doses of exercise interventions included are not stated, but these are important considerations when analyzing the effects of exercises. It is recommended to follow (PRISMA) guidelines and provide detailed inclusion criteria according to “PICO”.
  6. Table 3, the information on the exercise intervention should be included in this table. Figure 10-17 have labels of different types of exercise protocols (e.g., HIIT, MICT and aerobic), but there are no detailed explanations or information included in the method session.
  7. There was no discussion on the high level of heterogeneity. Potential reasons or mechanisms need to be provided to help the readers understand the results.

Author Response

Firstly, I would like to show my gratitude for taking time from you schedule and reviewing our manuscript. We are happy to see that you found our manuscript interesting.Regarding PROSPERO, the work has already been completed and we can no longer submit it.For your query regarding effects of the exercise on PO, I have elaborated para 5 of introduction which I hope resolves this issue. Secondly, our study is unique as it disregards any ambiguous results and provides more statistically significant results and focuses on effect of physical stress on cardiometabolic factors primarily body weight, BMI, diastolic blood pressure (DBP), and systolic blood pressure (SBP), HDL, LDL, blood glucose, and waist circumference. Thirdly, I have now provided a paragraph on sensitivity analysis in results sections and have given reason for high heterogeneity in our results. Lastly, we have updated the table and provided the modality of each article as well as formulated my inclusion/exclusion criteria based on PICOS.

Round 2

Reviewer 2 Report

Thank you for the opportunity to review a revision of “Effects of exercise in improving the cardiometabolic risk factors 3 in overweight children: a systematic review and meta-analysis.” While the authors’ revisions did address some of my previous comments (e.g., addition of PRISMA checklist and information). While the authors provide broad responses to my general comments, they did not provide individual responses/locations to my specific comments

Although additional information was added, the introduction still does not motivate the need for a review. The authors note that previous results were not consistent, but do not explain why a new review is needed and what this is adding.

Justification for search strategy is need needed.  In their response to my concern that articles could have been missed (as n = 12 seems low based on information provided in the manuscript) their reasons for exclusion of the two examples I provided were not clear to me in the exclusion/inclusion criteria section of the methods.  

The title suggests that “overweight” children are the focus of this review, but often throughout the manuscript, participants are referred to as obese. Overweight is not necessarily exchangeable with obese, and I don’t believe either were defined in the context of this review. 

The addition of the type of physical stress in Table 3 is helpful, but additional information such as dosage (frequency, session length, activity modalities) is important as well.

Author Response

Firstly, I would like to show my gratitude for taking time from you schedule and reviewing our manuscript. We are happy to see that you found our manuscript interesting. For your query regarding effects of the exercise on PO, I have elaborated para 5 of introduction which I hope resolves this issue. Secondly, our study is unique as it disregards any ambiguous results and provides more statistically significant results and focuses on effect of physical stress on cardiometabolic factors primarily body weight, BMI, diastolic blood pressure (DBP), and systolic blood pressure (SBP), HDL, LDL, blood glucose, and waist circumference. Thirdly, I have now provided a paragraph on sensitivity analysis in results sections and have given reason for high heterogeneity in our results. Lastly, we have updated the table and provided the modality of each article as well as formulated my inclusion/exclusion criteria based on PICOS.

For Introduction:

We have worked according to your line sequence and have changed and provide details regarding your queries. We have explained the word ‘multidisciplinary’ and what are the cardiometabolic factors we intend to work on. We have also specified the age and that we took overweight participants.

Queries answered line wise:

Line 31: We have clarified the sentence for better understanding.

Line 40: We have added a transition sentence.

Line 48: Both factors show influence and we have added effect of lifestyle as well.

Line 48: We have started with a new sentence now.

Line 51: We have added details of physical activity.

Line 58: We have discussed what is meant by “multidisciplinary” in the text.

Lines 59-64: We have given details of the multidisciplinary programs. 

Line 65: Cardiac metabolic risk factors were those that can be clinically worked upon and then they are used to analyze if these factors may cause increase in adiposity or blood pressure which can be brought down using stress intervention

Line 66: We have mentioned the participants are those with age less than 18 years so it includes both children and adolescents 

For Methods:

We have provided PRISMA checklist at the end of manuscript (after references) as directed by you. We assure you that we have not missed any articles as are inclusion/exclusion criteria with our search string allowed us to obtain all possible and relevant studies. Staiano et al must be excluded as it does not have follow-up mean ans standard deviation (it has baseline values and post intervention mean difference which we can’t use in our method of analysis). Leeuwena et al has used percentiles which is also not part of our method. We have updated our literature search criteria as per your instructions and we have stated the domains used by predefined criteria according to Cochrane Collaboration’s Tool. We have also described why we used Odds Ratio here.

Queries answered line wise:

Line 72: We have added prisma checklist and following are reasons to exclude the 2 studies suggested: Staiano et al must be excluded as it does not have follow-up mean ans standard deviation (it has baseline values and post intervention mean difference which we can’t use in our method of analysis). Leeuwena et al has used percentiles which is also not part of our method.

Line 83 and 84: We have updated our literature search criteria as per your instructions.

Line 87: We have added the different domains that were used in assessing bias.

Line 89: Description of why we used Odds Ratio has been provided.

For Results:

We have added details in the portion you highlighted for us and updated the table to provide the modalities of the given studies.

Queries answered line wise:

Line 115-116: We have now described what baseline and what controls were taken.

Lines 118-119: Comparison was made between the pre and post intervention values/findings.

Table 3: We have added another column with all the relevant data available about the interventions used

For Discussion:

We have added age criteria and stated that participants were overweight. We have also provided more discussion so that it becomes clearer about the importance of BMI in clinical practice.

Queries answered line wise:

Line 196: Age is now defined

Line 207: We have described the use of “overweight” in our study.

Line 219: We have taken your advice into account and changed the text accordingly.